# On the Curved Geometry of Accelerated Optimization

**Aaron Defazio**
Facebook AI Research
New York

## Abstract

In this work we propose a differential geometric motivation for Nesterov's accelerated gradient method (AGM) for strongly-convex problems. By considering the optimization procedure as occurring on a Riemannian manifold with a natural structure, The AGM method can be seen as the proximal point method applied in this curved space. This viewpoint can also be extended to the continuous time case, where the accelerated gradient method arises from the natural block-implicit Euler discretization of an ODE on the manifold. We provide an analysis of the convergence rate of this ODE for quadratic objectives.

## 1 Introduction

The core algorithms of convex optimization are gradient descent (GD) and the accelerated gradient method (AGM). These methods are rarely used directly, more often they occur as the building blocks for distributed, composite, or non-convex optimization. In order to build upon these components, it is helpful to understand not just *how* they work, but *why*. The gradient method is well understood in this sense. It is commonly viewed as following a direction of steepest descent or as minimizing a quadratic upper bound. These interpretations provide a motivation for the method as well as suggesting a potential convergence proof strategy.

The accelerated gradient method in contrast has an identity crisis. Its equational form is remarkably malleable, allowing for many different ways of writing the same updates. We list a number of these forms in Table 1. Nesterov's original motivation for the AGM method used the concept of estimate sequences. Unfortunately, estimate sequences do not necessarily yield the simplest accelerated methods when generalized, such as for the composite case (Beck and Teboulle 2009, Nesterov 2007), and they have not been successfully applied in the important finite-sum (variance reduced) optimization setting.

Because of the complexity of estimate sequences, the AGM method is commonly motivated as a form of momentum. This view is flawed as a way of introducing the AGM method from first principles, as the most natural way of using momentum yields the heavy ball method instead:

$$x^{k+1} = x^k - \gamma \nabla f\left(x^k\right) + \beta\left(x^k - x^{k-1}\right),$$

which arises from discretizing the physics of a particle in a potential well with additional friction. The heavy-ball method does not achieve an accelerated convergence rate on general convex problems, suggesting that momentum, *per se*, is not the reason for acceleration. Another contemporary view is the linear-coupling interpretation of Allen-Zhu and Orecchia [2017], which views the AGM method as an interpolation between gradient descent and mirror descent. We take a more geometric view in our interpretation.

In this work we motivate the AGM by introducing it as an application of the proximal-point method:

$$x^k = \arg\min_x \left\{ f(x) + \frac{\eta}{2} \left\| x - x^{k-1} \right\|^2 \right\}.$$

The proximal point (PP) method is perhaps as foundational as the gradient descent method, although it sees even less direct use as each step requires solving a regularized subproblem, in contrast to the closed form steps for GD and AGM. The PP method gains remarkable convergence rate properties in exchange for the computational difficulty, including convergence for any positive step-size.

We construct the AGM by applying a dual form of the proximal point method in a curved space. Each step follows a geodesic on a manifold in a sense we make precise in Section 4. We use the term curved with respect to a coordinate system, rather than a coordinate free notion of curvature such as the Riemannian curvature. We first give a brief introduction to the concepts from differential geometry necessary to understand our motivation. The equational form that our argument yields is much closer to those that have been successfully applied in practice, particularly for the minimization of finite sums [Lan and Zhou, 2017, Zhang and Xiao, 2017].

## 2    Connections

A *n (affine) connection* is a type of structure on a manifold that can be used to define and compute geodesics. Geodesics in this sense represent curves of zero acceleration. These geodesics are more general concepts than Riemannian geodesics induced by the Riemannian connection, not necessarily representing the shortest path in any metric. Indeed, we will define multiple connections on the same manifold that lead to completely different geodesics.

Given a $n$ dimensional coordinate system, a connection is defined by $n^3$ numbers at every point $x$ on the manifold, called the connection coefficients (or Christoffel symbols) $\Gamma_{ij}^k(x)$. A geodesic is a path $\gamma : [0,1] \to \mathcal{M}$ (in our case $\mathcal{M} = \mathbb{R}^n$) between two points $x$ and $y$ can then be computed as the unique solution $\gamma(t) = x(t)$ to the system of ordinary differential equations [Lee, 1997, Page 58, Eq 4.11]:

$$\frac{d^2\gamma^i}{dt^2} \doteq \frac{d^2 x^i}{dt^2} + \sum_{j,k} \Gamma_{jk}^i(x) \frac{dx^j}{dt} \frac{dx^k}{dt} = 0,$$

with boundary conditions $x(0) = x$ and $x(1) = y$. Here $x^i$ denotes the $i$th component of $x$ expressed in the same coordinate system as the connection.

## 3    Divergences induce Hessian manifold structure

Let $\phi$ be a smooth strongly convex function defined on $\mathbb{R}^n$. The Bregman divergence generated by $\phi$:

$$B_\phi(x,y) = \phi(x) - \phi(y) - \langle \nabla\phi(y), x - y \rangle,$$

and its derivatives can be used to define a remarkable amount of structure on the domain of $\phi$. In particular, we can define a *Riemannian manifold, together with two dually flat connections with biorthogonal coordinate* systems [Amari and Nagaoka, 2000, Shima, 2007]. This structure is also known as a Hessian manifold. Topologically it is $\mathcal{M} = \mathbb{R}^n$ with the following additional geometric structures.

### Riemannian structure

Riemannian manifolds have the additional structure of a *metric tensor* (a generalized dot-product), defined on their tangent spaces. We denote the vector space of tangent vectors at a point $x$ as $T_x\mathcal{M}$. If we express the tangent vectors with respect to the Euclidean basis, the metric at a point $x$ is a quadratic form with the Hessian matrix $H(x) = \nabla_x^2 B(x,y) = \nabla^2\phi(x)$ of $\phi$ at $x$:

$$g_x(u,v) = u^T H(x) v.$$

### Biorthogonal coordinate systems

Central to the notion of a manifold is the invariance to the choice of coordinate system. We can express a point on the manifold as well as a point in the tangent space using any coordinate system that is most convenient. Of course, when we wish to perform calculations on the manifold we must be careful to express all quantities in that coordinate system. Euclidean coordinates $e_i$ are the most natural on our Hessian manifold, however there is another coordinate system which is naturally dual to $e_i$, and ties the manifold structure directly to duality theory in optimization.

Table 1: Equivalent forms of Nesterov's method for $\mu$-strongly convex, $L$-smooth $f$. Proofs of the stated relations are available in the appendix.

| Form Name | Algorithm | Relations |
|---|---|---|
| Nesterov [2013] form I | $$y^k = \frac{\alpha\gamma v^k + \gamma x^k}{\alpha\mu + \gamma}$$ $$x^{k+1} = y^k - \frac{1}{L}\nabla f(y^k),$$ $$v^{k+1} = (1-\alpha)v^k + \frac{\alpha\mu}{\gamma}y^k - \frac{\alpha}{\gamma}\nabla f(y^k)$$ | $$\alpha_{\text{Nes}} = \sqrt{\mu/L}$$ $$\gamma_{\text{Nes}} = \mu.$$ |
| Nesterov [2013] form II | $$x^{k+1} = y^k - \frac{1}{L}\nabla f(y^k),$$ $$y^{k+1} = x^{k+1} + \beta\left(x^{k+1} - x^k\right)$$ | $$\beta_{\text{Nes}} = \frac{\sqrt{L}-\sqrt{\mu}}{\sqrt{L}+\sqrt{\mu}}$$ |
| Sutskever et al. [2013] | $$p^{k+1} = \beta p^k - \frac{1}{L}\nabla f\left(x^k + \beta p^k\right),$$ $$x^{k+1} = x^k + p^{k+1}$$ | $$p_{\text{Sut}}^{k+1} = x_{\text{Nes}}^{k+1} - x_{\text{Nes}}^k,$$ $$y_{\text{Nes}}^k = x_{\text{Sut}}^k + \beta p_{\text{Sut}}^k.$$ |
| Modern Momentum[1] | $$p^{k+1} = \beta p^k + \nabla f(x^k),$$ $$x^{k+1} = x^k - \frac{1}{L}\left(\nabla f(x^k) + \beta p^{k+1}\right).$$ | $$x_{\text{mod}}^k = x_{\text{Sut}}^k + \beta p_{\text{Sut}}^k = y_{\text{Nes}}^k,$$ $$p_{\text{mod}}^k = -Lp_{\text{Sut}}^k.$$ |
| Auslender and Teboulle [2006] | $$y^k = (1-\theta)\hat{x}^k + \theta z^k,$$ $$z^{k+1} = z^k - \frac{\gamma}{\theta}\nabla f(y^k),$$ $$\hat{x}^k = (1-\theta)\hat{x}^k + \theta z^{k+1}.$$ | $$\theta_{\text{AT}} = 1 - \beta_{\text{Nes}},$$ $$\hat{x}_{\text{AT}}^k = x_{\text{Nes}}^k,$$ $$y_{\text{AT}}^k = y_{\text{Nes}}^k = x_{\text{mod}}^k,$$ $$\gamma_{\text{AT}} = 1/L.$$ |
| Lan and Zhou [2017] | $$\tilde{x}^k = \alpha(x^{k-1} - x^{k-2}) + x^{k-1},$$ $$\underline{x}^k = \frac{\tilde{x}^k + \tau\underline{x}^{k-1}}{1+\tau},$$ $$g^k = \nabla f(\underline{x}^k),$$ $$x^k = x^{k-1} - \frac{1}{\eta}g^k.$$ | $$x_{\text{Lan}}^k = z_{\text{AT}}^k,$$ $$\underline{x}_{\text{Lan}}^k = y_{\text{AT}}^k,$$ $$\eta_{\text{Lan}} = \frac{\gamma_{\text{AT}}}{\theta_{\text{AT}}},$$ $$\tau_{\text{Lan}} = \frac{1-\theta_{\text{AT}}}{\theta_{\text{AT}}},$$ $$\alpha_{\text{Lan}} = 1 - \theta_{\text{AT}}.$$ |

Recall that for a convex function $\phi$ we may define the convex conjugate $\phi^*(y) = \max_x \left\{\langle x, y \rangle - \phi(x)\right\}$. The dual coordinate system we define simply identifies each point $x$, when expressed in Euclidean ("primal") coordinates, with the vector of "dual" coordinates:

$$y = \nabla\phi(x).$$

Our assumptions of smoothness and strong convexity imply this is a one-to-one mapping, with inverse given by $x = \nabla\phi^*(y)$. The remarkable fact that the gradient of the conjugate is the inverse of the gradient is a key building block of the theory in this paper.

The notion of *biorthogonality* refers to natural tangent space coordinates of these two systems. A tangent vector $v$ at a point $x$ can be converted to a vector $u$ of dual (tangent space) coordinates using matrix multiplication with the Hessian [Shima, 2007]:

$$u = H(x)v, \tag{1}$$

Given the definition of the metric above, it is easy to see that if we have two vectors $v_1$ and $v_2$, we may express $v_2$ in dual coordinates $u_2$ so that the metric tensor takes the simple form:

$$g_x(v_1, v_2) = v_1^T H(x)v_2 = v_1^T H(x)\left(H(x)^{-1}u_2\right) = v_1^T u_2,$$

which is the *biorthogonal* relation between the two tangent space coordinate systems.

**Dual Flat Connections**

There is an obvious connection $\Gamma^{(E)}$ we can apply to the Hessian manifold, the Euclidean connection that trivially identifies straight lines in $\mathbb{R}^n$ as geodesics. Normally when we perform gradient descent

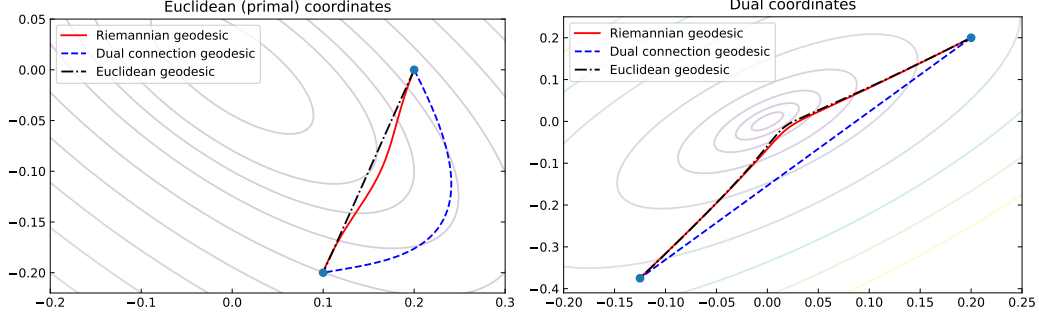

Figure 1: Illustrative geodesics for $f(x) = \frac{1}{4} \|Ax\|^4$, with $A = [2, 1; 1, 3]$. Viewing them from both coordinate systems highlights the duality. Contour lines are for $f$ and $f^*$ respectively.

in $\mathbb{R}^n$ we are implicitly following a geodesic of this connection. The connection coefficients $\Gamma_{ij}^{(E)k}$ are all zero when this connection is expressed in Euclidean coordinates. A connection that has $\Gamma_{ij}^k = 0$ with respect to some coordinate system is a *flat* connection.

The Hessian manifold admits another flat connection, which we will call the dual connection, as it corresponds to straight lines in the dual coordinate system established above. In particular each dual geodesic can be expressed in primal coordinates $\gamma(t)$ as a solution to the equation:

$$\nabla \phi \left( \gamma(t) \right) = at + b,$$

for vectors $a$, $b$ representing the initial velocity and point respectively (both represented in dual coordinates) that depend on the boundary conditions. This is quite easy to solve using the relation $\nabla \phi^{-1} = \nabla \phi^*$ discussed above. For instance, a geodesic $\gamma : [0, 1] \to \mathcal{M}$ between two arbitrary points $x$ and $y$ under the dual connection could be computed explicitly in Euclidean coordinates as:

$$\gamma(t) = \nabla \phi^* \left( t \nabla \phi(y) + (1 - t) \nabla \phi(x) \right).$$

If we instead know the initial velocity we can find the endpoint with:

$$y = \nabla \phi^* \left( \nabla \phi(x) + H(x^k)v \right). \tag{2}$$

The flatness of the dual connection $\Gamma^{(D)}$ is crucial to its computability in practice. If we instead try to compute the geodesic in Euclidean coordinates using the geodesic ODE, we have to work with the connection coefficients which involve third derivatives of $\phi$ (taking the form of double those of the Riemannian connection $\Gamma^{(R)}$):

$$\Gamma_{ij}^{(D)k}(x) = 2\Gamma_{ij}^{(R)k} = \left[ H(x)^{-1} \left( \nabla H(x) \right)_i \right]_{kj},$$

The Riemannian connection's geodesics are similarly difficult to compute directly from the ODE (they also can't generally be expressed in a simpler form).

## 4 Bregman proximal operators follow geodesics

Bregman divergences arise in optimization primarily through their use in proximal steps. A Bregman proximal operation balances finding a minimizer of a given function $f$ with maintaining proximity to a given point $y$, measured using a Bregman divergence instead of a distance metric:

$$x^k = \arg\min_x \left\{ f(x) + \rho B_\phi(x, x^{k-1}) \right\}. \tag{3}$$

A core application of this would be the mirror descent step [Nemirovski and Yudin, 1983, Beck and Teboulle, 2003], where the operation is applied to a linearized version of $f$ instead of $f$ directly:

$$x^k = \arg\min_x \left\{ \langle x, \nabla f(x^{k-1}) \rangle + \rho B_\phi(x, x^{k-1}) \right\}.$$

Bregman proximal operations can be interpreted as geodesic steps with respect to the dual connection. The key idea is that given an input point $x^{k-1}$, they output a point $x$ such that the velocity of the connecting geodesic is equal to $-\nabla \frac{1}{\rho} f(x)$ at $x$. This velocity is measured in the flat coordinate system of the connection, the dual coordinates. To see why, consider a geodesic $\gamma(t) = (1 -$

$t)\nabla\phi(x^{k-1}) + t\nabla\phi(x^k)$. Here $x^{k-1}$ and $x^k$ are in primal coordinates and $\gamma(t)$ is in dual coordinates. The velocity is $\frac{d}{dt}\gamma(t) = \nabla\phi(x^k) - \nabla\phi(x^{k-1})$. Contrast to the optimality condition of the Bregman prox (Equation 3):

$$-\frac{1}{\rho}\nabla f(x^k) = \nabla\phi(x^k) - \nabla\phi(x^{k-1}).$$

For instance, when using the Euclidean penalty the step is:

$$x^k = \arg\min_x\left\{f(x) + \frac{\rho}{2}\left\|x - x^{k-1}\right\|^2\right\}.$$

The final velocity is just $x^k - x^{k-1}$, and so $x^k - x^{k-1} = -\frac{1}{\rho}\nabla f(x^k)$, which is the solution of the proximal operation.

## 5  Primal-Dual form of the proximal point method

The proximal point method is the building block from which we will construct the accelerated gradient method. Consider the basic form of the proximal point method applied to a strongly convex function $f$. At each step, the iterate $x^k$ is constructed from $x^{k-1}$ by solving the proximal operation subproblem given an inverse step size parameter $\eta$:

$$x^k = \arg\min_x\left\{f(x) + \frac{\eta}{2}\left\|x - x^{k-1}\right\|^2\right\}. \tag{4}$$

This step can be considered an implicit form of the gradient step, where the gradient is evaluated at the end-point of the step instead of the beginning:

$$x^k = x^{k-1} - \frac{1}{\eta}\nabla f(x^k),$$

which is just the optimality condition of the subproblem in Equation 4, found by taking the derivative $\nabla f(x) + \eta x - \eta x^{k-1}$ to be zero. A remarkable property of the proximal operation becomes apparent when we rearrange this formula, namely that the solution to the operation is not a single point but a *primal-dual pair*, whose weighted sum is equal to the input point:

$$x^k + \frac{1}{\eta}\nabla f(x^k) = x^{k-1}.$$

If we define $g^k = \nabla f(x^k)$, the primal-dual pair obeys a duality relation: $g^k = \nabla f(x^k)$ and $x^k = \nabla f^*(g^k)$, which allows us to interchange primal and dual quantities freely. Indeed we may write the condition in a dual form as:

$$\nabla f^*\left(g^k\right) + \frac{1}{\eta}g^k = x^{k-1}, \tag{5}$$

which is the optimality condition for the proximal operation:

$$g^k = \arg\min_g\left\{f^*(g) + \frac{1}{2\eta}\left\|g - \eta x^{k-1}\right\|^2\right\}.$$

Our goal in this section is to express the proximal point method in terms of a dual step, and while this equation involves the dual function $f^*$, it is not a *step* in the sense that $g^k$ is formed by a proximal operation from $g^{k-1}$.

We can manipulate this formula further to get an update of the form we want, by simply adding and subtracting $g^{k-1}$ from 5:

$$\nabla f^*\left(g^k\right) + \frac{1}{\eta}g^k = \frac{1}{\eta}g^{k-1} + \left(x^{k-1} - \frac{1}{\eta}g^{k-1}\right),$$

Which gives the updates:

$$g^k = \arg\min_g\left\{f^*(g) - \left\langle g,\, x^{k-1} - \frac{1}{\eta}g^{k-1}\right\rangle + \frac{1}{2\eta}\left\|g - g^{k-1}\right\|^2\right\},$$

$$x^k = x^{k-1} - \frac{1}{\eta}g^k.$$

We call this the primal-dual form of the proximal point method.

# 6   Acceleration as a change of geometry

The proximal point method is rarely used in practice due to the difficulty of computing the solution to the proximal subproblem. It is natural then to consider modifications of the subproblem to make it more tractable. The subproblem becomes particularly simple if we replace the proximal operation with a Bregman proximal operation with respect to $f^*$,

$$g^k = \arg\min_g \left\{ f^*(g) - \left\langle g,\, x^{k-1} - \frac{1}{\eta}g^{k-1} \right\rangle + \tau B_{f^*}(g, g^{k-1}) \right\}.$$

We have additionally changed the penalty parameter to a new constant $\tau$, which is necessary as the change to the Bregman divergence changes the scaling of distances. We discuss this further below.

Recall from Section 4 that Bregman proximal operations follow geodesics. The key idea is that we are now following a geodesic in the dual connection of $\phi = f^*$, using the notation of Section 3, which is a *straight-line in the primal coordinates* of $f$ due to the flatness of the connection (Section 3). Due to the flatness property, a simple closed-form solution can be derived by equating the derivative to 0:

$$\nabla f^*(g^k) - \left[ x^{k-1} - \frac{1}{\eta}g^{k-1} \right] + \tau\nabla f^*(g^k) - \tau\nabla f^*(g^{k-1}) = 0,$$

$$\text{therefore } g^k = \nabla f\left( (1+\tau)^{-1}\left[ x^{k-1} - \frac{1}{\eta}g^{k-1} + \tau\nabla f^*(g^{k-1}) \right] \right).$$

This formula gives $g^k$ in terms of the derivative of known quantities, as $\nabla f^*(g^{k-1})$ is known from the previous step as the point at which we evaluated the derivative at. We will denote this argument to the derivative operation $y$, so that $g^k = \nabla f(y^k)$. It no longer holds that $g^k = \nabla f(x^k)$ after the change of divergence. Using this relation, $y$ can be computed each step via the update:

$$y^k = \frac{x^{k-1} - \frac{1}{\eta}g^{k-1} + \tau y^{k-1}}{1+\tau}.$$

In order to match the accelerated gradient method exactly we need some additional flexibility in the step size used in the $y^k$ update. To this end we introduce an additional constant $\alpha$ in front of $g^{k-1}$, which is 1 for the proximal point variant. The full method is as follows:

---

**Bregman form of the accelerated gradient method**

$$y^k = \frac{x^{k-1} - \frac{\alpha}{\eta}g^{k-1} + \tau y^{k-1}}{1+\tau},$$

$$g^k = \nabla f(y^k),$$

$$x^k = x^{k-1} - \frac{1}{\eta}g^k. \tag{6}$$

---

This is very close to the equational form of Nesterov's method explored by Lan and Zhou [2017], with the change that they assume an explicit regularizer is used, whereas we assume strong convexity of $f$. Indeed we have chosen our notation so that the constants match. This form is algebraically equivalent to other known forms of the accelerated gradient method for appropriate choice of constants. Table 1 shows the direct relation between the many known ways of writing the accelerated gradient method in the strongly-convex case (Proofs of these relations are in the Appendix). When $f$ is $\mu$-strongly convex and $L$-smooth, existing theory implies an accelerated geometric convergence rate of at least $1 - \sqrt{\frac{\mu}{L}}$ for the parameter settings [Nesterov, 2013]:

$$\eta = \sqrt{\mu L}, \qquad \tau = \frac{L}{\eta}, \qquad \alpha = \frac{\tau}{1+\tau}.$$

In contrast, the primal-dual form of the proximal point method achieves at least that convergence rate for parameters:

$$\eta = \sqrt{\mu L}, \qquad \tau = \frac{1}{\eta}, \qquad \alpha = 1.$$

The difference in $\tau$ arises from the difference in the scaling of the Bregman penalty compared to the Euclidean penalty. The Bregman generator $f^*$ is strongly convex with constant $1/L$ whereas the Euclidean generator $\frac{1}{2}\left\|\cdot\right\|^2$ is strongly convex with constant 1, so the change in scale requires rescaling by $L$.

## 6.1 Interpretations

After the change in geometry, the $g$ update no longer gives a dual point that is directly the gradient of the primal iterate. However, notice that the term we are attempting to minimize in the $g$ step:

$$f^*(g) - \langle g, x^{k-1} - \frac{\alpha}{\eta} g^{k-1} \rangle,$$

has a fixed point of $\nabla f^*(g^k) = x^{k-1} - \frac{\alpha}{\eta} g^k$, which is precisely an $\alpha$-weight version of the proximal point's key property from Equation 5. Essentially we have relaxed the proximal-point method. Instead of this relation holding precisely at every step, we are instead constantly taking steps which pull $g$ closer to satisfying it.

## 6.2 Inertial form

The primal-dual view of the proximal point method can also be written in terms of the quantity $z^{k-1} = x^{k-1} - \frac{\alpha}{\eta} g^{k-1}$ instead of $x^{k-1}$. This form is useful for the construction of ODEs that model the discrete dynamics. Under this change of variables the updates are:

$$g^k = \arg\min_g \left\{ f^*(g) - \langle g, z^{k-1} \rangle + \frac{1}{2\eta} \|g - g^{k-1}\|^2 \right\},$$

$$z^k = z^{k-1} - \frac{1}{\eta} g^k - \frac{\alpha}{\eta} (g^k - g^{k-1}). \tag{7}$$

## 6.3 Relation to the heavy ball method

Consider Equation 6 with $\alpha = 0$, which removes the over-extrapolation before the proximal operation. If we define $\beta = \frac{\tau}{1+\tau}$ we may write the method as:

$$x^k = x^{k-1} - \frac{1}{\eta} f'(y^{k-1}), \qquad y^k = \beta y^{k-1} + (1-\beta) x^k.$$

We can eliminate $x^k$ from the $y^k$ update above by plugging in the $x^k$ step equation, then using the $y^k$ update from the previous step in the form $(1-\beta) x^{k-1} = y^{k-1} - \beta y^{k-2}$:

$$y^k = \beta y^{k-1} + (1-\beta) \left( x^{k-1} - \frac{1}{\eta} f'(y^{k-1}) \right)$$

$$= \beta y^{k-1} - (1-\beta) \frac{1}{\eta} f'(y^{k-1}) + [y^{k-1} - \beta y^{k-2}]$$

$$= y^{k-1} - (1-\beta) \frac{1}{\eta} f'(y^{k-1}) + \beta [y^{k-1} - y^{k-2}].$$

This has the exact form of the heavy ball method with step size $(1-\beta)/\eta$ and momentum $\beta$. We can also derive the heavy ball method by starting from the saddle-point expression for $f$:

$$\min_x f(x) = \min_x \max_g \left\{ \langle x, g \rangle - f^*(g) \right\}.$$

The alternating-block gradient descent/ascent method on the objective $\langle x, g \rangle - f^*(g)$ with step-size $\gamma$ is simply:

$$g^k = g^{k-1} + \frac{1}{\gamma} \left[ x^{k-1} - \nabla f^*(g^{k-1}) \right], \qquad x^k = x^{k-1} - \gamma g^k.$$

If we instead perform a Bregman proximal update in the dual geometry for the $g$ part, we arrive at the same equations as we had for the primal-dual proximal point method but with $\alpha = 0$, yielding the heavy ball method. In order to get the accelerated gradient method instead of the heavy ball method, the extra inertia that arises from starting from the proximal point method instead of the saddle point formulation appears to be crucial.

## 7 Dual geometry in continuous time

The inertial form (Equation 7) of the proximal point method can be formulated as an ODE in a very natural way, by mapping $z^k - z^{k-1} \to \dot{z}$ and $g^k - g^{k-1} \to \dot{g}$, and taking $x$ and $g$ to be at time $t$.

This is the inverse of the Euler class of discretizations applied separately to the two terms, which is the most natural way to discretize an ODE. The resulting proximal point ODE is:

$$\dot{g} = f_g(z, g, t) \doteq -\frac{1}{\tau}\nabla f^*(g) + \frac{1}{\tau}z,$$

$$\dot{z} = f_z(z, g, t) \doteq -\frac{1}{\eta}g - \frac{\alpha}{\eta}\dot{g}.$$

We have suppressed the dependence on $t$ of each quantity for notational simplicity. We can treat $g$ more formally as a point $g \in \mathcal{M}$ on a Hessian manifold $\mathcal{M}$. Then the solution for the $g$ variable of the ODE is a curve $\gamma(t) : I \rightarrow \mathcal{TM}$ from an interval $I$ to the tangent bundle on the manifold so the velocity $\dot{\gamma}(t) \in T_g\mathcal{M}$ obeys the ODE: $\dot{\gamma}(t) = f_g(z, g, t)$. The right hand side of the ODE is a point in the tangent space of the manifold at $\gamma(t)$, expressed in Euclidean coordinates.

We can now apply the same partial change of geometry that we used in the discrete case. We will consider the quantity $f_g(z, g, t)$ to be a tangent vector in dual tangent space coordinates For the $\phi = f^*$ Hessian manifold, instead of its primal tangent space coordinates (which would leave the ODE unchanged). The variable $g$ remains in primal coordinates with respect to $\phi$, so we must add to the ODE a change of coordinates for the tangent vector, yielding:

$$\dot{g} = \left(\nabla^2 f^*(g)\right)^{-1} f_g(z, g, t),$$

where we have used the inverse of Equation 1, with $\phi = f^*$. We can rewrite this as:

$$f_g(z, g, t) = \nabla^2 f^*(g)\dot{g} = \frac{d}{dt}\nabla f^*(g),$$

giving the AGM ODE system:

$$\frac{d}{dt}\nabla f^*(g) = -\frac{1}{\tau}\nabla f^*(g) + \frac{1}{\tau}z, \qquad \dot{z} = -\frac{1}{\eta}g - \frac{\alpha}{\eta}\dot{g}.$$

It is now easily seen that the implicit Euler update for the $g$ variable with $z$ fixed now corresponds to the solution of the Bregman proximal operation considered in the discrete case. So this ODE is a natural continuous time analogue to the accelerated gradient method.

**Convergence in continuous time**

The natural analogy to convergence in continuous time is known as the decay rate of the ODE. A sufficient condition for an ODE with parameters $u = [z; g]$ to decay with constant $\rho$ is:

$$\|u(t) - u^*\| \le \exp\left(-t\rho\right)\|u(0) - u^*\|,$$

where $u^*$ is a fixed point. We can relate this to the discrete case by noting that $\exp(-t\rho) = \lim_{k \to \infty}(1 - \frac{t}{k}\rho)^k$, so given our discrete-time convergence rate is proportional to $(1 - \sqrt{\mu/L})^k$, we would expect values of $\rho$ proportional to $\sqrt{\mu/L}$ if the ODE behaves similarly to the discrete process. We have been able to establish this result for both the proximal and AGM ODEs for quadratic objectives (proof in the Appendix in the supplementary material).

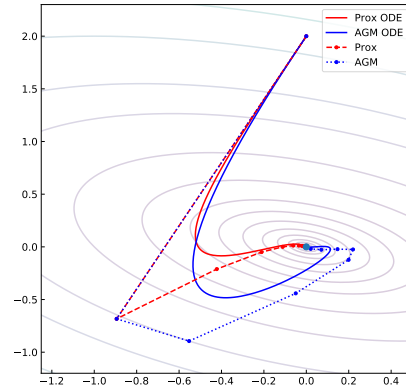

Figure 2: Paths for the quadratic problem $f(x) = \frac{1}{2}x^T A x$ with $A = [2, 1; 1, 3]$.

**Theorem 1.** *The proximal and AGM ODEs decay with at least the following rates for $\mu$-strongly convex and $L$-smooth quadratic objective functions when using the same hyper-parameters as in the discrete case:*

$$\rho_{prox} \ge \frac{\sqrt{\mu}}{\sqrt{\mu}+\sqrt{L}}, \quad \rho_{AGM} \ge \frac{1}{2}\sqrt{\frac{\mu}{L}}.$$

Figure 2 contrasts the convergence of the discrete and continuous variants. The two methods have quite distinct paths whose shape is shared by their ODE counterparts.

## 8 Related Work

The application of Bregman divergence to the analysis of continuous time views of the accelerated gradient method has recently been explored by Wibisono et al. [2016] and Wilson et al. [2018]. Their approaches do not use the Bregman divergence of $f^*$, a key factor of our approach. The Bregman divergence of a function $\phi$ occurs explicitly as a term in a Hamiltonian, in contrast to our view of $\phi$ as curving space. The accelerated gradient method has been shown to be modeled by a momentum of the form ODE $\ddot{X} + c(t)\dot{X} + \nabla f(x) = 0$ by Su et al. [2014]. Natural discretizations of their ODE result in the heavy-ball method instead of the accelerated gradient method, unlike our form which can produce both based on the choice of $\alpha$. The fine-grained properties of momentum ODEs have also been studied in the quadratic case by Scieur et al. [2017].

A primal-dual form of the regularized accelerated gradient method appears in Lan and Zhou [2017]. Our form can be seen as a special case of their form when the regularizer is zero. Our work extends theirs, providing an understanding of the role that geometry plays in unifying acceleration and implicit steps.

The Riemannian connection induced by a function has been heavily explored in the optimization literature as part of the natural gradient method [Amari, 1998], although other connections on this manifold are less explored. The dual-flat connections have primarily seen use in the information-geometry setting for optimization over distributions [Amari and Nagaoka, 2000].

The accelerated gradient method is not the only way to achieve accelerated rates among first order methods. Other techniques include the Geometric descent method of Bubeck et al. [2015], where a bounding ball is updated at each step that encloses two other balls, a very different approach. The method described by Nemirovski and Yudin [1983] is also notable as being closer to the conjugate gradient method than other accelerated approaches, but at the expense of requiring a 2D search instead of a 1D line search at each step.

## 9 Conclusion

We believe the tools of differential geometry may provide a new and insightful avenue for the analysis of accelerated optimization methods. The analysis we provide in this work is a first step in this direction. The advantage of the differential geometric approach is that it provides high level tools that make the derivation of acceleration easier to state. This derivation, from the proximal point method to the accelerated gradient method, is in our opinion not nearly as mysterious as the other known approaches to understanding acceleration.

## Footnotes

[1]PyTorch & Tensorflow (for instance) implement this form. Evaluating the gradient and function at the current iterate $x^k$ instead of a shifted point makes it more consistent with gradient descent when wrapped in a generic optimization layer.

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
