[Supplementary Material · curved_geom2019_full.pdf]

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

}$ <br> $x^{k+1} = y^k - \dfrac{1}{L}\nabla f(y^k),$ <br> $v^{k+1} = (1-\alpha)v^k + \dfrac{\alpha\mu}{\gamma}y^k - \dfrac{\alpha}{\gamma}\nabla f(y^k)$ | $\alpha_{\text{Nes}} = \sqrt{\mu/L}$ <br> $\gamma_{\text{Nes}} = \mu.$ |
| Nesterov [2013] form II | $x^{k+1} = y^k - \dfrac{1}{L}\nabla f(y^k),$ <br> $y^{k+1} = x^{k+1} + \beta\left(x^{k+1} - x^k\right)$ | $\beta_{\text{Nes}} = \dfrac{\sqrt{L}-\sqrt{\mu}}{\sqrt{L}+\sqrt{\mu}}$ |
| Sutskever et al. [2013] | $p^{k+1} = \beta p^k - \dfrac{1}{L}\nabla f\left(x^k + \beta p^k\right),$ <br> $x^{k+1} = x^k + p^{k+1}$ | $p_{\text{Sut}}^{k+1} = x_{\text{Nes}}^{k+1} - x_{\text{Nes}}^k,$ <br> $y_{\text{Nes}}^k = x_{\text{Sut}}^k + \beta p_{\text{Sut}}^k.$ |
| Modern Momentum[1] | $p^{k+1} = \beta p^k + \nabla f(x^k),$ <br> $x^{k+1} = x^k - \dfrac{1}{L}\left(\nabla f(x^k) + \beta p^{k+1}\right).$ | $x_{\text{mod}}^k = x_{\text{Sut}}^k + \beta p_{\text{Sut}}^k = y_{\text{Nes}}^k,$ <br> $p_{\text{mod}}^k = -L p_{\text{Sut}}^k.$ |
| Auslender and Teboulle [2006] | $y^k = (1-\theta)\hat{x}^k + \theta z^k,$ <br> $z^{k+1} = z^k - \dfrac{\gamma}{\theta}\nabla f(y^k),$ <br> $\hat{x}^k = (1-\theta)\hat{x}^k + \theta z^{k+1}.$ | $\theta_{\text{AT}} = 1 - \beta_{\text{Nes}},$ <br> $\hat{x}_{\text{AT}}^k = x_{\text{Nes}}^k,$ <br> $y_{\text{AT}}^k = y_{\text{Nes}}^k = x_{\text{mod}}^k,$ <br> $\gamma_{\text{AT}} = 1/L.$ |
| Lan and Zhou [2017] | $\tilde{x}^k = \alpha(x^{k-1} - x^{k-2}) + x^{k-1},$ <br> $\underline{x}^k = \dfrac{\tilde{x}^k + \tau\underline{x}^{k-1}}{1+\tau},$ <br> $g^k = \nabla f(\underline{x}^k),$ <br> $x^k = x^{k-1} - \dfrac{1}{\eta}g^k.$ | $x_{\text{Lan}}^k = z_{\text{AT}}^k,$ <br> $\underline{x}_{\text{Lan}}^k = y_{\text{AT}}^k,$ <br> $\eta_{\text{Lan}} = \dfrac{\gamma_{\text{AT}}}{\theta_{\text{AT}}},$ <br> $\tau_{\text{Lan}} = \dfrac{1-\theta_{\text{AT}}}{\theta_{\text{AT}}},$ <br> $\

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

# Appendix

## A  Reformulations of the accelerated gradient method

**Form II**

This simplification is described in Nesterov [2013] which we reproduce for completeness. Recall that Form I is given by the updates:

$$y^k = \frac{\alpha\gamma v^k + \gamma x^k}{\alpha\mu + \gamma},$$

$$x^{k+1} = y^k - \frac{1}{L}\nabla f(y^k),$$

$$v^{k+1} = (1 - \alpha)\, v^k + \frac{\alpha\mu}{\gamma}y^k - \frac{\alpha}{\gamma}\nabla f(y^k).$$

Nesterov specifies the requirement that $\gamma = (1-\alpha)\gamma + \alpha\mu$. When $\alpha = \sqrt{\mu/L}$ then $\gamma$ must then satisfy:

$$\gamma = (1 - \sqrt{\mu/L})\gamma + \mu\sqrt{\mu/L},$$

$$\therefore 1 = (1 - \sqrt{\mu/L}) + \frac{\mu}{\gamma}\sqrt{\mu/L},$$

$$\therefore \sqrt{\mu/L}) = \frac{\mu}{\gamma}\sqrt{\mu/L},$$

$$\therefore \mu = \gamma$$

We may rewrite the $y^k$ definition as:

$$(\alpha\mu + \gamma)\, y^k = \alpha\gamma v^k + \gamma x^k,$$

$$\therefore v^k = \frac{1}{\alpha\gamma}\left[(\alpha\mu + \gamma)\, y^k - \gamma x^k\right].$$

Plugging this into the $v$ step:

$$
\begin{aligned}
v_{k+1} &= (1 - \alpha)\, v^k + \frac{\alpha\mu}{\gamma}y^k - \frac{\alpha}{\gamma}\nabla f(y^k) \\
&= \frac{1 - \alpha}{\alpha\gamma}\left[(\alpha\mu + \gamma)\, y^k - \gamma x^k\right] + \frac{\alpha\mu}{\gamma}y^k - \frac{\alpha}{\gamma}\nabla f(y^k) \\
&= \frac{1}{\gamma}\left[(1 - \alpha)\,\mu y^k + \left(\frac{1-\alpha}{\alpha}\right)\gamma y^k + \alpha\mu y^k\right] - \frac{1 - \alpha}{\alpha}x^k - \frac{\alpha}{\gamma}\nabla f(y^k) \\
&= \frac{1}{\alpha\gamma}\left[\alpha\mu y^k + (1 - \alpha)\,\gamma y^k\right] - \frac{1 - \alpha}{\alpha}x^k - \frac{\alpha}{\gamma}\nabla f(y^k) \\
&= \frac{1}{\alpha}\left[y^k\right] - \frac{1 - \alpha}{\alpha}x^k - \frac{\alpha}{\gamma}\nabla f(y^k) \\
&= x^k + \frac{1}{\alpha}\left(y^k - x^k\right) - \frac{\alpha}{\gamma}\nabla f(y^k) \\
&= x^k + \frac{1}{\alpha}\left(x^{k+1} - x^k\right).
\end{aligned}
$$

For $y^k$, we start with using $\gamma = \mu$, then apply the $v$ simplification;

$$
\begin{aligned}
y^k &= \frac{\alpha\gamma v^k + \gamma x^k}{\alpha\gamma + \gamma} \\
&= \frac{\left(\alpha\gamma x^{k-1} + \gamma\left(x^k - x^{k-1}\right)\right) + \gamma x^k}{\alpha\gamma + \gamma} \\
&= x^k + \frac{\left(\alpha\gamma x^{k-1} + \gamma\left(x^k - x^{k-1}\right)\right) - \alpha\gamma x^k}{\alpha\gamma + \gamma} \\
&= x^k + \frac{\alpha\gamma\left(x^{k-1} - x^k\right) + \gamma\left(x^k - x^{k-1}\right)}{\alpha\gamma + \gamma} \\
&= x^k + \frac{\gamma - \alpha\gamma}{\alpha\gamma + \gamma}\left(x^k - x^{k-1}\right).
\end{aligned}
$$

Note that by multiplying by $\sqrt{L}/\mu$ we get:

$$
\frac{\mu - \mu\sqrt{\mu/L}}{\mu\sqrt{\mu/L} + \mu} = \frac{\sqrt{L} - \sqrt{\mu}}{\sqrt{\mu} + \sqrt{L}} = \beta.
$$

**Sutskever's form**

Recall Sutskever's form:

$$
\begin{aligned}
p_{\text{Sut}}^{k+1} &= \beta p_{\text{Sut}}^k - \frac{1}{L}\nabla f\left(x_{\text{Sut}}^k + \beta p_{\text{Sut}}^k\right), \\
x_{\text{Sut}}^{k+1} &= x_{\text{Sut}}^k + p_{\text{Sut}}^{k+1}.
\end{aligned}
$$

and Nesterov's form:

$$
\begin{aligned}
x_{\text{Nes}}^{k+1} &= y_{\text{Nes}}^k - \frac{1}{L}\nabla f(y_{\text{Nes}}^k), \\
y_{\text{Nes}}^{k+1} &= x_{\text{Nes}}^{k+1} + \beta\left(x_{\text{Nes}}^{k+1} - x_{\text{Nes}}^k\right).
\end{aligned}
$$

We will show that using the substitutions:

$$
\begin{aligned}
p_{\text{Sut}}^{k+1} &= x_{\text{Nes}}^{k+1} - x_{\text{Nes}}^k, \\
y_{\text{Nes}}^k &= x_{\text{Sut}}^k + \beta p_{\text{Sut}}^k,
\end{aligned}
$$

applied to Sutskever's form gives Nesterov's form. We start with the momentum term:

$$
p_{\text{Sut}}^{k+1} = \beta p_{\text{Sut}}^k - \frac{1}{L}\nabla f\left(y_{\text{Nes}}^k\right),
$$

$$
\therefore x_{\text{Nes}}^{k+1} - x_{\text{Nes}}^k = \beta\left(x_{\text{Nes}}^k - x_{\text{Nes}}^{k-1}\right) - \frac{1}{L}\nabla f\left(y_{\text{Nes}}^k\right),
$$

$$
\therefore x_{\text{Nes}}^{k+1} + \frac{1}{L}\nabla f\left(y_{\text{Nes}}^k\right) = x_{\text{Nes}}^k + \beta\left(x_{\text{Nes}}^k - x_{\text{Nes}}^{k-1}\right).
$$

Defining: $y_{\text{Nes}}^k := x_{\text{Nes}}^{k+1} + \frac{1}{L}\nabla f\left(y_{\text{Nes}}^k\right)$ and applying on the right gives Nesterov's $y$ update:

$$
y_{\text{Nes}}^k = x_{\text{Nes}}^k + \beta\left(x_{\text{Nes}}^k - x_{\text{Nes}}^{k-1}\right).
$$

**Modern form**

We want:

$$
\begin{aligned}
p_{\text{Mod}}^{k+1} &= \beta p_{\text{Mod}}^k + \nabla f(x_{\text{Mod}}^k), \\
x_{\text{Mod}}^{k+1} &= x_{\text{Mod}}^k - \frac{1}{L}\left(\nabla f(x_{\text{Mod}}^k) + \beta p_{\text{Mod}}^{k+1}\right).
\end{aligned}
$$

Starting from Sutskever's form,

$$
\begin{aligned}
p_{\text{Sut}}^{k+1} &= \beta p_{\text{Sut}}^k - \frac{1}{L}\nabla f\left(x_{\text{Sut}}^k + \beta p_{\text{Sut}}^k\right), \\
x_{\text{Sut}}^{k+1} &= x_{\text{Sut}}^k + p_{\text{Sut}}^{k+1}.
\end{aligned}
$$

Define $x_{\text{Mod}}^k = x_{\text{Sut}}^k + \beta p_{\text{Sut}}^k$. Note that this is equal to $y_{\text{Nes}}^k$ by definition. So we have:

$$x_{\text{Sut}}^k = x_{\text{Mod}}^k - \beta p_{\text{Sut}}^k.$$

Plugging that into the Sutskever step

$$x_{\text{Mod}}^{k+1} - \beta p_{\text{Sut}}^{k+1} = x_{\text{Mod}}^k - \beta p_{\text{Sut}}^k + p_{\text{Sut}}^{k+1},$$

$$\therefore x_{\text{Mod}}^{k+1} = x_{\text{Mod}}^k + \left(p_{\text{Sut}}^{k+1} - \beta p_{\text{Sut}}^k\right) + \beta p_{\text{Sut}}^{k+1}$$

$$= x_{\text{Mod}}^k - \frac{1}{L}f'\left(x_{\text{Mod}}^k\right) + \beta p_{\text{Sut}}^{k+1}.$$

Then define $p_{\text{mod}}^k = -Lp_{\text{Sut}}^k$, so the update becomes:

$$x_{\text{Mod}}^{k+1} = x_{\text{Mod}}^k - \frac{1}{L}\left(\nabla f\left(x_{\text{Mod}}^k\right) + \beta p_{\text{Sut}}^{k+1}\right).$$

The momentum update changes from:

$$p_{\text{Sut}}^{k+1} = \beta p_{\text{Sut}}^k - \frac{1}{L}\nabla f\left(x_{\text{Sut}}^k + \beta p_{\text{Sut}}^k\right),$$

to:

$$-\frac{1}{L}p_{\text{mod}}^{k+1} = -\beta\frac{1}{L}p_{\text{mod}}^k - \frac{1}{L}\nabla f\left(x_{\text{Mod}}^k\right),$$

$$\therefore p_{\text{mod}}^{k+1} = \beta p_{\text{mod}}^k + \nabla f\left(x_{\text{Mod}}^k\right).$$

**Auslender & Teboulle form**

$$y_{\textbf{AT}}^k = (1 - \theta_{\textbf{AT}})\hat{x}_{\textbf{AT}}^k + \theta_{\textbf{AT}}z_{\textbf{AT}}^k,$$

$$z_{\textbf{AT}}^{k+1} = z_{\textbf{AT}}^k - \frac{\gamma_{\textbf{AT}}}{\theta_{\textbf{AT}}}\nabla f(y_{\textbf{AT}}^k),$$

$$\hat{x}_{\textbf{AT}}^{k+1} = (1 - \theta_{\textbf{AT}})\hat{x}_{\textbf{AT}}^k + \theta_{\textbf{AT}}z_{\textbf{AT}}^{k+1}.$$

We first eliminate $y_{\text{AT}}^k$ from the $\hat{x}_{\text{AT}}^{k+1}$ update:

$$\hat{x}_{\text{AT}}^{k+1} = (1 - \theta_{\text{AT}})\hat{x}_{\text{AT}}^k + \theta_{\text{AT}}z_{\text{AT}}^{k+1}$$

$$= (1 - \theta_{\text{AT}})\hat{x}_{\text{AT}}^k + \theta_{\text{AT}}\left(z_{\text{AT}}^k - \frac{\gamma_{\text{AT}}}{\theta_{\text{AT}}}\nabla f(y_{\text{AT}}^k)\right)$$

$$= \hat{x}_k + \theta_{\text{AT}}\left(z_{\text{AT}}^k - \hat{x}_{\text{AT}}^k\right) - \gamma_{\text{AT}}\nabla f\left(\hat{x}_k + \theta_{\text{AT}}\left(z_{\text{AT}}^k - \hat{x}_{\text{AT}}^k\right)\right).$$

This notational similarity with Nesterov's $x_{\text{Nes}}^{k+1} = y_{\text{Nes}}^k - \frac{1}{L}\nabla f(y_{\text{Nes}}^k)$ suggests matching

$$y_{\text{Nes}}^k = x_{\text{mod}}^k = y_{\text{AT}}^k = \hat{x}_k + \theta_{\text{AT}}\left(z_{\text{AT}}^k - \hat{x}_{\text{AT}}^k\right),$$

as well as $\gamma_{\text{AT}} = \frac{1}{L}$ and

$$x_{\text{Nes}}^k = \hat{x}_{\text{AT}}^k.$$

Note that using using the step for $\hat{x}$ we can rearrange to get:

$$z_{\text{AT}}^{k+1} = \frac{1}{\theta_{\text{AT}}}\hat{x}_{\text{AT}}^{k+1} - \frac{(1 - \theta_{\text{AT}})}{\theta_{\text{AT}}}\hat{x}_{\text{AT}}^k.$$

Now to determine $\theta$ we simplify using this substitution:

$$y_{\text{AT}}^k = \hat{x}_{\text{AT}}^k + \theta_{\text{AT}}\left(z_{\text{AT}}^k - \hat{x}_{\text{AT}}^k\right)$$

$$= \hat{x}_{\text{AT}}^k + \theta_{\text{AT}}\left(\frac{1}{\theta_{\text{AT}}}\hat{x}_{\text{AT}}^k - \frac{(1 - \theta_{\text{AT}})}{\theta_{\text{AT}}}\hat{x}_{\text{AT}}^{k-1}\right)$$

$$= \hat{x}_{\text{AT}}^k + \theta_{\text{AT}}\left(\frac{1 - \theta_{\text{AT}}}{\theta_{\text{AT}}}\hat{x}_{\text{AT}}^k - \frac{(1 - \theta_{\text{AT}})}{\theta_{\text{AT}}}\hat{x}_{\text{AT}}^{k-1}\right)$$

$$= \hat{x}_{\text{AT}}^k + (1 - \theta_{\text{AT}})\left(\hat{x}_{\text{AT}}^k - \hat{x}_{\text{AT}}^{k-1}\right).$$

So my matching constants against Nesterov's method we have $\beta_{\text{Nes}} = 1 - \theta_{\text{AT}}$.

**Lan form**

$$\tilde{x}_{\text{Lan}}^{k} = \alpha_{\text{Lan}}(x_{\text{Lan}}^{k-1} - x_{\text{Lan}}^{k-2}) + x_{\text{Lan}}^{k-1},$$

$$\underline{x}_{\text{Lan}}^{k} = \frac{\tilde{x}_{\text{Lan}}^{k} + \tau_{\text{Lan}}\underline{x}_{\text{Lan}}^{k-1}}{1 + \tau_{\text{Lan}}},$$

$$g_{\text{Lan}}^{k} = \nabla f(\underline{x}_{\text{Lan}}^{k}),$$

$$x_{\text{Lan}}^{k} = x_{\text{Lan}}^{k-1} - \frac{1}{\eta_{\text{Lan}}}g_{\text{Lan}}^{k}.$$

Now we can eliminate $\tilde{x}$ from the updates to give

$$\underline{x}_{\text{Lan}}^{k} = \frac{\tau_{\text{Lan}}\underline{x}_{\text{Lan}}^{k-1} + x_{\text{Lan}}^{k-1} + \alpha_{\text{Lan}}(x_{\text{Lan}}^{k-1} - x_{\text{Lan}}^{k-2})}{1 + \tau_{\text{Lan}}},$$

Also consider the $y^k$ update for the AT method:

$$y_{\textbf{AT}}^{k} = (1 - \theta_{\textbf{AT}})\hat{x}_{\textbf{AT}}^{k} + \theta_{\textbf{AT}}z_{\textbf{AT}}^{k},$$

We can write this to not involve $\hat{x}$, giving:

$$y_{\text{AT}}^{k} = (1 - \theta_{\text{AT}})y_{\text{AT}}^{k-1} + \theta_{\text{AT}}z_{AT}^{k} + (1 - \theta_{\text{AT}})\theta_{\text{AT}}\left(z_{\text{AT}}^{k} - z_{\text{AT}}^{k-1}\right).$$

This form suggests matching the iterates with:

$$\underline{x}_{\text{Lan}}^{k} = y_{\text{AT}}^{k}, \quad x_{\text{Lan}}^{k} = z_{\text{AT}}^{k}.$$

Under this matching, the constants need to satisfy the following relations:

$$(1 - \theta_{\text{AT}}) = \frac{\tau_{\text{Lan}}}{1 + \tau_{\text{Lan}}},$$

$$\left(\theta_{\text{AT}} + \theta_{\text{AT}} - \theta_{\text{AT}}^{2}\right) = \frac{1 + \alpha_{\text{Lan}}}{1 + \tau_{\text{Lan}}},$$

$$(1 - \theta_{\text{AT}})\theta_{\text{AT}} = \frac{\alpha_{\text{Lan}}}{1 + \tau_{\text{Lan}}}.$$

The settings:

$$\tau_{\text{Lan}} = \frac{1 - \theta_{\text{AT}}}{\theta_{\text{AT}}}, \quad \alpha_{\text{Lan}} = 1 - \theta_{\text{AT}},$$

result in these three constraints being satisfied:

$$\frac{\tau}{1 + \tau} = \frac{\frac{1-\theta}{\theta}}{1 + \frac{1-\theta}{\theta}} = \frac{1 - \theta}{\theta + 1 - \theta} = 1 - \theta \checkmark,$$

$$\frac{1 + \alpha}{1 + \tau} = \frac{1 + 1 - \theta}{1 + \frac{1-\theta}{\theta}} = \frac{2\theta - \theta^2}{\theta + 1 - \theta} = 2\theta - \theta^2 \checkmark,$$

$$\frac{\alpha}{1 + \tau} = \frac{1 - \theta}{1 + \frac{1-\theta}{\theta}} = \frac{\theta - \theta^2}{1} \checkmark.$$

Matching the $x_{\text{Lan}}^{k}$ step $x_{\text{Lan}}^{k} = x_{\text{Lan}}^{k-1} - \frac{1}{\eta_{\text{Lan}}}g_{\text{Lan}}^{k}$. requires $\frac{1}{\eta_{\text{Lan}}} = \frac{\gamma_{\text{AT}}}{\theta_{\text{AT}}}$ also.

# B  Continuous time theory

**Lemma 2.** *Consider a linear ODE of the form:*

$$\dot{u} = A\left(u - u^{*}\right),$$

*Where $A$ is real and diagonalizable but not necessarily normal or symmetric, and $x^{*}$ is the fixed point. Then*

$$\|u(t) - u^{*}\| \leq \max_{j} \exp\left(tRe[\lambda_{j}]\right)\|u(0) - u^{*}\|,$$

*where $\lambda_{j}$ are eigenvalues of $A$ with real part $Re[\lambda_{j}]$.*

*Proof.* Because the inhomogeneous term $Au^*$ is constant w.r.t time, without loss of generality we can reduce our problem to a homogenous ODE by shifting the origin:

$$\dot{u} = Au.$$

The solution of a linear ODE of this form is given in closed form using the matrix exponential:

$$u(t) = \exp\left(tA\right)u(0),$$

we can use this formula to bound the norm of $u(t)$:

$$\therefore \|u(t)\| \leq \|\exp\left(tA\right)\| \|u(0)\|,$$

where $\|\cdot\|$ is the spectral norm. If $A$ is diagonalizable with $A = U\mathrm{diag}(\lambda_1, \ldots \lambda, d)U^{-1}$ then the matrix exponential can be expressed as:

$$\exp\left(tA\right) = U\left[\begin{array}{ccc} \exp(t\lambda_1) & 0 & 0 \\ 0 & \ddots & 0 \\ 0 & 0 & \exp(t\lambda_d) \end{array}\right]U^{-1}.$$

The spectral norm is given by the largest absolute value of the eigenvalues, so we must consider the interaction of the real and complex parts. For an eigenvalue $\lambda = a + bi$ of $A$, the norm takes the simple form:

$$\begin{aligned} |\exp(t\lambda)| &= \exp\left(ta + tbi\right) \\ &= |\exp\left(ta\right)| |\exp\left(tbi\right)| \\ &= |\exp\left(ta\right)|. \end{aligned}$$

So the spectral norm is given by the maximum over the eigenvalues $\lambda_j$ of $\exp\left(tRe[\lambda_j]\right)$. $\qquad\square$

**Theorem 3.** *Consider the following linear ODE:*

$$\dot{u} = A\left(u - u^*\right),$$

$$A : 2n \times 2n = \left[\begin{array}{cc} -I & -\frac{1}{\eta}I + H^{-1} \\ \eta I & -\eta H^{-1} \end{array}\right],$$

*where $H : n \times n$ is a real, positive definite and symmetric matrix with minimum eigenvalue $\mu$ and maximum eigenvalue $L$. This corresponds to the proximal ODE for a quadratic function $f(x) = \frac{1}{2}\left(x - x^*\right)^T H\left(x - x^*\right)$, with $u = [x; g]$ and $u^* = [x^*; 0]$. Then the decay rate of the ODE towards the origin $u^* = 0$ can be bounded as follows for $\eta = \sqrt{\mu L}$:*

$$\|u(t) - u^*\| \leq \exp\left(-t\rho\right)\|u(0) - u^*\|,$$

$$\rho = \frac{\sqrt{\mu}}{\sqrt{\mu} + \sqrt{L}}.$$

*Proof.* We will take the approach of bounding the real parts of the eigenvalues of $A$, so that we can directly apply Lemma 2.

Let $U\Lambda U^T = H$ be the eigen-decomposition of $H$. Note that the operation of conjugation by $U$ leaves the identity matrix unchanged ($UIU^T = I$). Each block of A is just a weighted combination of the identity matrix and $H^{-1}$, so this implies that conjugation of a block by it's self gives a diagonal matrix. We can use this idea to define a similarity transform that converts $A$ into a matrix where each of the four blocks are diagonal. In particular we have:

$$\left[\begin{array}{cc} U & 0 \\ 0 & U \end{array}\right]^T\left[\begin{array}{cc} A_{11} & A_{12} \\ A_{21} & A_{22} \end{array}\right]\left[\begin{array}{cc} U & 0 \\ 0 & U \end{array}\right] = \left[\begin{array}{cc} UA_{11}U^T & UA_{12}U^T \\ UA_{21}U^T & UA_{22}U^T \end{array}\right],$$

where we have written A in terms of its 4 constituent $n \times n$ blocks. Each block is a weighted sum of the identity matrix and the diagonal matrix of inverse eigenvalues of H, for instance:

$$UA_{12}U^T = -\frac{1}{\eta}I + \Lambda^{-1}.$$

Next we construct a permutation matrix $\Pi : 2n \times 2n$ with the goal of converting $D$ into a block diagonal matrix with $2 \times 2$ blocks along the diagonal, where each block has the structure of $A$ as if it was applied to a 1D optimization problem, with $H$ being replaced by one of the $d$ eigenvalues of $H$. This is achieved with the permutation matrix $\Pi$ that is zero except for:

$$\Pi_{2i,i} = 1, \; \Pi_{2i+1,d+i} = 1, \; i = 1 \ldots n.$$

For instance, in the $n = 2$ case the matrix is:

$$\begin{bmatrix} 1 & 0 & 0 & 0 \\ 0 & 0 & 1 & 0 \\ 0 & 1 & 0 & 0 \\ 0 & 0 & 0 & 1 \end{bmatrix}.$$

This matrix has the effect of interleaving the primal dual pairs (per coordinate) instead of having all the primal coordinates together followed by all the dual coordinates. So when we conjugate using $\Pi$ we get:

$$\Pi^T \begin{bmatrix} U & 0 \\ 0 & U \end{bmatrix}^T A \begin{bmatrix} U & 0 \\ 0 & U \end{bmatrix} \Pi = \begin{bmatrix} T_1 & 0 & \cdots \\ 0 & \ddots & 0 \\ \vdots & 0 & T_d \end{bmatrix},$$

where each $T$ is a $2 \times 2$ matrix of the described form:

$$T_i = \begin{bmatrix} -1 & -\frac{1}{\eta} + \lambda_i^{-1} \\ \eta & -\eta\lambda_i^{-1} \end{bmatrix}.$$

The eigenvalues of a block diagonal matrix are just the eigenvalues of the blocks concatenated, and since there is a similarity transform between $A$ and this block diagonal matrix, we have effectively reduced our problem to considering 1D quadratics, with curvature between $\mu$ and $L$, for fixed $\eta$.

Recall that for a matrix $\begin{bmatrix} a & b \\ c & d \end{bmatrix}$ the eigenvalues are given by the two roots of a quadratic, namely:

$$\nu_\pm = \frac{a + d \pm \sqrt{(a+d)^2 - 4(ad - bc)}}{2}.$$

We use the notation $\nu$ to avoid confusion between the eigenvalues of the $T$ blocks and those of H . For a block $T_i$, this expression is

$$\nu_+^{(i)} = -\frac{1}{2} - \frac{\eta\lambda_i^{-1}}{2} \pm \frac{1}{2}\sqrt{\left(1 + \eta\lambda_i^{-1}\right)^2 - 4\left(\eta\lambda_i^{-1} + 1 - \eta\lambda_i^{-1}\right)}$$

Suppose that the discriminate (the quantity under the square root) is negative, then

$$Re\left[\nu_\pm^{(i)}\right] = -\frac{1}{2} - \frac{\eta\lambda_i^{-1}}{2},$$

this is obviously at least as small as $-\rho = -\frac{\sqrt{\mu}}{\sqrt{\mu} + \sqrt{L}}$, since the largest value of $\rho$ possible is when $\mu = L$, in which case $\rho = \frac{1}{2}$. So consider instead the case where the discriminate is positive. We need only consider the $v_+$ root as it is strictly larger. Then we will use the concavity of the square root function to bound $\nu_+$,

$$h(x) \le h(y) + \langle \nabla h(y), x - y \rangle$$

For $h = \sqrt{\cdot}$, with $y = \left(-1 - \eta\lambda_i^{-1}\right)^2$ and $x = \left(-1 - \eta\lambda_i^{-1}\right)^2 - 4$. We get:

$$\nu_+^{(i)} \le -\frac{1}{2} - \frac{\eta\lambda_i^{-1}}{2} + \frac{1}{2}\sqrt{\left(1 + \eta\lambda_i^{-1}\right)^2} - \frac{4}{4\left(1 + \eta\lambda_i^{-1}\right)}$$

$$= -\frac{1}{\left(1 + \eta\lambda_i^{-1}\right)}.$$

Therefore:

$$\nu_+^{(i)} \le -\frac{1}{\left(1 + \eta\lambda_i^{-1}\right)} \le -\frac{1}{\left(1 + \eta/\mu\right)} = -\frac{\sqrt{\mu}}{\sqrt{\mu} + \sqrt{L}} = -\rho.$$

Thus we have shown that the real parts of all eigenvalues of $A$ are less than $-\rho$. $\qquad\square$

**Theorem 4.** *Consider the following linear ODE:*

$$\dot{u} = A\left(u - u^*\right),$$

$$A : 2d \times 2d = \begin{bmatrix} -\frac{\alpha}{\eta\tau}H & -\frac{1}{\eta}I + \frac{\alpha}{\eta\tau}I \\ \frac{1}{\tau}H & -\frac{1}{\tau}I \end{bmatrix},$$

*where $H : d \times d$ is a real, positive definite and symmetric matrix with minimum eigenvalue $\mu$ and maximum eigenvalue $L$. This corresponds to the AGM ODE for a quadratic objective $f(x) = \frac{1}{2}(x - x^*)^T H (x - x^*)$, with $u = [x; g]$ and $u^* = [x^*; 0]$. The decay rate of this ODE towards the origin $u^* = 0$ can be bounded as follows for $\eta = \sqrt{\mu L}$, $\tau = L/\eta$, and $\alpha \in [0, 1]$:*

$$\|u(t)\| \leq \exp\left(-t\rho\right)\|u(0)\|,$$

$$\rho = \frac{1}{2}\sqrt{\frac{\mu}{L}}.$$

*Proof.* We can apply the same proof technique as for Theorem 3, we omit the details. The $2 \times 2$ block diagonal matrices are:

$$T_i = \begin{bmatrix} -\frac{\alpha}{\eta\tau}\lambda_i & -\frac{1}{\eta} + \frac{\alpha}{\eta\tau} \\ \frac{1}{\tau}\lambda_i & -\frac{1}{\tau} \end{bmatrix}.$$

For eigenvalues $\lambda_i$ of $H$. The two eigenvalues of $T_i$ are:

$$\nu_\pm^{(i)} = -\frac{\alpha}{2\eta\tau}\lambda_i - \frac{1}{2\tau} \pm \frac{1}{2}\sqrt{\left(\frac{\alpha}{\eta\tau}\lambda_i + \frac{1}{\tau}\right)^2 - 4\frac{\lambda_i}{\eta\tau}}$$

The discriminate here is always non-positive. To see why, we plug in the constants $\eta$ and $\tau$:

$$\left(\frac{\alpha}{L}\lambda_i + \sqrt{\frac{\mu}{L}}\right)^2 - 4\frac{\lambda_i}{L} \leq 2\frac{\alpha^2}{L^2}\lambda_i^2 + 2\frac{\mu}{L} - 4\frac{\lambda_i}{L}$$

$$\leq 2\frac{\lambda_i}{L} + 2\frac{\mu}{L} - 4\frac{\lambda_i}{L}$$

$$= 2\frac{\mu}{L} - 2\frac{\lambda_i}{L}$$

$$\leq 0.$$

So the real part is given by the quantity outside the square root, namely:

$$Re\left[\nu_\pm^{(i)}\right] = -\frac{\alpha}{2\eta\tau}\lambda_i - \frac{1}{2\tau} = -\frac{\alpha\lambda_i}{2L} - \frac{1}{2}\sqrt{\frac{\mu}{L}} \leq -\frac{1}{2}\sqrt{\frac{\mu}{L}}.$$

Using Lemma 2 gives the result. □

## C The standard heavy ball ODE

The standard heavy-ball ODE for a quadratic $f(x) = \frac{1}{2}x^T H x$ is

$$\ddot{x} + (1 - \beta)\dot{x} + \gamma H x = 0.$$

Which can be written in first-order form in terms of a momentum parameter $p$ as:

$$\dot{x} = p,$$
$$\dot{p} = -(1 - \beta)p - \gamma H x. \tag{8}$$

The constants that result in optimal convergence rate for the discretize heavy ball method:

$$x^{k+1} = x^k + p^k$$
$$p^k = \beta p^{k-1} - \gamma \nabla f(x^k)$$

which can also be written as:

$$x^{k+1} = x^k - \gamma H x^k + \beta \left( x^k - x^{k-1} \right),$$

are:

$$\beta = \frac{\sqrt{L} - \sqrt{\mu}}{\sqrt{L} + \sqrt{\mu}}, \quad \gamma = \frac{4}{\left( \sqrt{L} + \sqrt{\mu} \right)^2}. \tag{9}$$

**Theorem 5.** *Consider the following ODE:*

$$\dot{u} = A \left( u - u^* \right),$$

$$A = \begin{bmatrix} 0 & I \\ -\gamma H & -\left( 1 - \beta \right) I \end{bmatrix}.$$

*This is the ODE in Equation 8 written in matrix form for a combined iterate u. For the parameters given in Equation 9, this ODE has decay rate at least:*

$$\| u(t) - u^* \| \leq \exp \left( -t\rho \right) \| u(0) - u^* \|,$$

$$\text{where } \rho = \frac{\sqrt{\mu}}{\sqrt{L} + \sqrt{\mu}}.$$

*Proof.* We can reduce the problem to considering the eigenvalues of $2 \times 2$ matrices as we did for the proximal and AGM ODEs. We have matrices of the form:

$$T_i = \begin{bmatrix} 0 & 1 \\ -\gamma \lambda_i & -\left( 1 - \beta \right) \end{bmatrix},$$

whose eigenvalues are given by the general formula

$$\nu_\pm = \frac{a + d \pm \sqrt{(a+d)^2 - 4(ad - bc)}}{2}.$$

Simplifying:

$$\nu_\pm = -\frac{1}{2} \left( 1 - \beta \right) \pm \frac{1}{2} \sqrt{\left( 1 - \beta \right)^2 - 4\gamma \lambda_i}.$$

Note that:

$$1 - \beta = 2 \frac{\sqrt{\mu}}{\sqrt{L} + \sqrt{\mu}}.$$

The choice of step size ensures that the discriminant is always non-positive:

$$\left( \frac{2\sqrt{\mu}}{\sqrt{L} + \sqrt{\mu}} \right)^2 - 4 \frac{4\lambda_i}{\left( \sqrt{L} + \sqrt{\mu} \right)^2} \leq \left( \frac{2\sqrt{\mu}}{\sqrt{L} + \sqrt{\mu}} \right)^2 - 4 \left( \frac{2\sqrt{\mu}}{\sqrt{L} + \sqrt{\mu}} \right)^2 \leq 0$$

Therefore the decay rate is bounded by the real part of the eigenvalues, which is $-\frac{1}{2}(1 - \beta)$. □