[Reviews · NeurIPS 2019]

Reviewer 1



Somewhere in Section 2, clarify what $\mathcal{M}$ is. I suppose it is simply R^n, with its usual manifold structure. Eq. between lines 54 and 55: Perhaps use a different notation to disambiguate between the acceleration of (the components of) \gamma (which is with respect to the affine connection), and the acceleration of the components of x, which seems to be in the classical sense. For example, in the book by Absil, Mahony and Sepulchre, they use D/dt for covariant derivatives, and d/dt for classical derivatives. So, for acceleration induced by the affine connection, we would have D^2/dt^2. Section 3: For smooth functions, strict convexity does not necessarily imply that the Hessian is everywhere positive definite (though the reverse is true). Isn't this an issue when using the Hessian of $\phi$ to define a Riemannian metric? Around equation 1: Can you explain more how the two discussions under "biorthogonal coordinate systems" are related? My understanding is: if we perturb x along the direction v, then y is perturbed along the direction D(\nabla \phi)(x)[v] = Hess \phi(x)[v] = H(x)v, and we call this u. If this is the right way to think about it, I would suggest to include it. When thinking about these two structures, I am wondering: would it help to think of two manifolds, M and N, and to think of \nabla \phi and \nabla \phi* as a diffeomorphism between these two manifolds? Then, the relationship between tangent spaces is clear through this map, and each manifold can have its own flat connection. I am more familiar with the definition of connection as a map which takes two smooth vector fields and outputs one smooth vector field, corresponding to a directional derivative of one along the other. If available, can you give a reference for a textbook that shows it is equivalent to specify the geodesics? Line 116: the velocity should be a vector; -f(x)/\rho seems to be missing \nabla. Also, it should be specified that this is the velocity in dual coordinates (on a similar note: the sentence goes on to say "... of the connection", but it would be good to specify which connection explicitly. Section 5: Here, phi was set to a special form (spell it out?) so the Bregman divergence is just the squared norm. But then, duality is used here by applying to f the concepts that were exposed for phi, regardless of Bregman divergence concerns. Specifically, for any smooth, strictly convex function q, we have a well defined q*, and the gradients of q and q* are inverse maps of one another, giving a diffeomorphism for R^n. This is used in two ways in the paper: it's used with phi in the context of Bregman divergences; and it's used with f in the context of primal-dual proximal point methods. Is this interpretation correct? Perhaps this could be clarified? Lines 138-140: Is the point that you would like an expression for g^k that the distance between $g$ and $g^{k-1}$ rather than $x^{k-1}$? This could be stated more directly. Line 154, "due to the flatness property, a simple closed-form solution can be derived by equating the derivative to 0" -- why is flatness necessary here? The definition of g^k at the bottom of page 5 is a the unique solution to a striclty convex problem; there's no need for flatness to claim that the optimum is attained when the gradient wrt g is zero? How do we initialize the algorithm, equation (5)? Do we need to be able to compute at least the very first "gradient of the conjugate of f"? Is that an issue in practice? Line 211: wouldn't \dot \gamma(t) be in the tangent space to the tangent bundle of M, rather than in the tangent space to the manifold M itself? Line 20: simpliest -> simplest Line 45: is type -> is a type Line 51: $x$on -> $x$ on Line 52: mention the name "Christoffel symbols"? Line 78: missing comma "coordinate, with" Line 130: missing reference after "Equation" (also, "i.e." should not start a sentence) Line 215: it's -> its

Reviewer 2



Starting from the well-know paper [2] about the explanation of acceleration, this area has been developed for several years. For this area, I have two criteria to measure the importance: 1) A good explanation should explain a large class of acceleration phenomenons. Nesterov's acceleration can be used for many settings, such as convex/strongly convex, composite/non-composite, first-order/high-order, etc. If we can explain various settings, then our explanation is less possible to be "overfitting". Inspired by estimation sequence, [1] proposed a unified theory for first-order algorithms to explain nearly all the settings of first-order algorithms, and showed substantial importance and elegance of the estimation sequence technique. 2) A good explanation should be able to give some new results and insights. Although the explanation itself is important, its practical value is its impact of developing new algorithms. As examples, [2] and [3] propose some restart heuristics according to the explanation. Unfortunately, the reviewer found that this paper can not satisfy the two criteria properly. As a unified theory has been given, this paper can only address the strongly convex setting. Meanwhile, I can not find something new induced by this explanation. The differential geometry perspective is novel. However, the dual linearity of Bregman iteration is a common sense in this community. The concept of flat connection is just another description of dual linearity. When the concept of differential geometry is used, we always expect something different from Euclidean space. If the author can show that acceleration can still work for "nontrivial" connections, this paper will have impacts. However, the authors seem only using the concept in differential geometry and not exploring it in depth. The introduction section is not state of the art and seems misleading. As [1] showed, the modern formulation of estimation sequence has strong power to explain acceleration. [1] Diakonikolas, Jelena, and Lorenzo Orecchia. "The approximate duality gap technique: A unified theory of first-order methods." SIAM Journal on Optimization 29.1 (2019): 660-689. [2] Su, Weijie, Stephen Boyd, and Emmanuel Candes. "A differential equation for modeling Nesterov’s accelerated gradient method: Theory and insights." Advances in Neural Information Processing Systems. 2014. [3] Krichene, Walid, Alexandre Bayen, and Peter L. Bartlett. "Accelerated mirror descent in continuous and discrete time." Advances in neural information processing systems. 2015.

Reviewer 3



The paper explains well its connections to related work. I would like to see the authors' view of how the geometric descent method of [1] fits into the picture. The authors should also say more about the difference between their method ond that of Allen-Zhu and Orecchia 2017; isn't the distinction not the geometry, but motivating via the proximal method? It seems the authors have squeezed the space from above certain display equations. I consider this a grave sin. Certainly other authors also had plenty to say, and figured out how to say it in 8 pages, following the template. There are sections you could remove without much loss. For example, the heavy ball section, 6.3, adds little. I still found the definitions of biorthogonality and of flatness of a connection confusing. I am not sure whether you can explain these better, or whether these ideas are just not suited to an 8 page format. I hope you can explain them better. Why is the relation above line 87 called "biorthogonal", and why is it important? It seems like a simple change of variables. As for flat connections, can you say why the connection coefficients vanish for the dual connection? A reference back to the equation below line 54 might also help. I struggled in particular to extract meaning from lines 102-105. [1] Sebastian Bubeck, Yin Tat Lee, Mohit Singh. "A geometric alternative to Nesterov’s accelerated gradient descent" https://arxiv.org/pdf/1506.08187.pdf Detailed comments: (I'm picky with you on grammar here because it's so close to perfect. Also an expository paper should have perfect grammar.) * capitalization errors in abstract, also line 30 * spacing error line 51 * the spacing above the eqn below line 68 is absurdly compressed. * line 116: have you left a \nabla off here? (to match the display equation below) * line 120: sentence fragment. * line 134: comma splice. * line 163: why do you think your method has three parameters? shouldn't two be enough? * below line 197: should be a nabla, not prime * line 215: its, not it's * line 216: write out wrt * line 218: sentence fragment * line 245: generally one avoids contractions in formal writing unless it serves a higher purpose. * line 248: why does momentum yield the accelerated method for Su 2014, but not for you? * Bibliography needs cleaning. You have two references to the same article by Su, Boyd, and Candes!

[Author Response · NeurIPS 2019]

**Reviewer 1**    Thank you for the comments on how to improve the notation, and the specified typos. We will incorporate these changes. Our responses to your main questions are below.

- We will restrict our discussion to strongly convex functions rather than strictly convex functions in the differential geometry section, as we don't currently handle the extra complexity from strict convexity in our discussion. Thank you for pointing this out.

- The view of the gradient map as a diffeomorphism is a useful way of visualizing what's going on, we will mention this in the updated draft. We will provide the reference regarding geodesics that you request as well.

- We do apply the Bregman divergence machinery to both $\phi$ and $f$. $\phi$ is used in the introductory sections rather and $f$ throughout because we technically use the conjugate of $f$, not $f$ its self, in the main change of geometry, and we wished to avoid notation involving conjugate-of-conjugate operations. We will update the wording to make this clearer.

- At line 154, the flatness ensures the solution has a simple form, you are correct that a solution will always exist even without flatness.

**Reviewer 2**    - The research avenue of understanding acceleration by discretizing continuous time dynamics is certainly one of the leading approaches currently being pursued. We have followed this literature carefully, as and far as we are aware all the existing approaches suffer from an "ad-hoc"-ary problem, where going from the continuous to discrete form requires a non-obvious or non-standard discretization method. The Approximate duality gap technique suffers from this problem, where as they state "... we can introduce an additional gradient step whose role is to cancel out the discretization error by reducing the upper bound.". Although certainly a promising approach, we don't consider this a satisfying explanation for the performance of acceleration, since it would be difficult to arrive at Nesterov's method via this path without already being aware of its functional form (i.e. working backwards). In contrast, our technique just relies on replacing the Bregman divergence used in the proximal point method with another more easily computable divergence, which is a natural step.

"As [1] showed, the modern formulation of estimation sequence has strong power to explain acceleration."' - We don't believe that the mentioned paper refutes our statement about estimate sequences, as they only derive composite-dual averaging via their discretization technique, not the simpler Beck and Teboulle method, which is the explicit claim we make. Likewise they do not apply their technique to variance reduced objectives, which is problematic to address using continuous time dynamics. The accelerated form of Lan that we build upon does apply to variance reduced problems.

Addressing point (2), our technique does lead to a ODE whose forward Euler discretization is naturally a discrete accelerated method, unlike the approach in [1]. We believe the explanation that our method provides, i.e. that Nesterov's method is the proximal point method in disguise, provides substantial insight that other explanations lack.

We hope that Reviewer 2 will revise his judgement based on our comments, although we acknowledge that the utility of new explanations of existing algorithms is subjective.

**Reviewer 3**    "To me, the weakest aspect of the paper is that the new interpretation of the accelerated gradient method still does not explain why it achieves acceleration" This is certainly the fundamental question. The high-level approach we attempt in this work is to show that it is just a form of the proximal point method. We believe the proximal point method is "intuitively" fast, but of course this is subjective.

- The mentioned Bubeck et. al. paper is perhaps the most unique approach to acceleration developed in recent years. We don't discuss it in detail in our work as the approach of finding points in the intersection of two balls is quite distinct from the proximal approach we look at, and it is not equivalent to Nesterov's method.

- The linear coupling paper by Allen-Zhu and Orecchia provides an interpretation of acceleration as linearly interpolating between a primal and a mirror step. This interpretation uses an equational form of acceleration similar to the one we build upon. It's certainly another valid interpretation, but we believe it lacks explanatory power, as it is unclear *why* such an interpolation would give an accelerated rate. As they state, it requires a substantial amount of analysis to derive the interpolation constant, which is not the case in our version, where the $\tau$ factor is directly given by the change-of-geometry.

- The heavy ball section could potentially be cut. We kept it in as we were often receiving questions about the relation to the heavy ball method whenever we discussed our work with others. We will move it to the appendix.

- You are correct that the biorthogonal condition is just a change of variables for the tangent spaces. It's notable just because of it's particular simplicity for these two coordinate systems.

Thank you for the remaining detailed comments. We do not have space here to address them one-by-one but we will update the manuscript to fix all the mentioned issues and provide clarification where requested.

[Meta-Review · NeurIPS 2019]

This work presents a nice link between acceleration and proximal point method, reminiscent of Catalyst (which could be cited) but in a simpler way. As proximal point methods and approximations, such as extragradient, are rarely used, research in this direction is interesting. Please update the manuscript according to the reviewers' comments for the final version.